# Validation of the Standardized Needs Evaluation Questionnaire in Polish Cancer Patients

**DOI:** 10.3390/cancers16081451

**Published:** 2024-04-09

**Authors:** Karolina Osowiecka, Anna Dolińska, Marek Szwiec, Eliza Działach, Jacek J. Nowakowski, Monika Rucińska

**Affiliations:** 1Department of Psychology and Sociology of Health and Public Health, School of Public Health, University of Warmia and Mazury in Olsztyn, Warszawska 30, 10-082 Olsztyn, Poland; 2Psychology Outpatient Clinic, University Hospital in Zielona Gora, Zyty 26, 65-046 Zielona Gora, Poland; annadolinska85@gmail.com; 3Department of Surgery and Oncology, Faculty of Medicine and Health Sciences, University of Zielona Gora, Zyty 28, 65-046 Zielona Gora, Poland; m.szwiec@cm.uz.zgora.pl; 4Department of Public Health, Faculty of Sciences in Bytom, Medical University of Silesia in Katowice, Piekarska 18, 41-902 Bytom, Poland; eliza.dzialach@gmail.com; 5Department of Botany and Evolutionary Ecology, Faculty of Biology and Biotechnology, University of Warmia and Mazury in Olsztyn, Plack Łódzki 3, 10-727 Olsztyn, Poland; jacek.nowakowski@uwm.edu.pl; 6Department of Oncology, Collegium Medicum University of Warmia and Mazury in Olsztyn, Wojska Polskiego 37, 10-228 Olsztyn, Poland; m_rucinska@poczta.onet.pl

**Keywords:** cancer patients, needs evaluation questionnaire, non-medical needs, unmet needs, validation

## Abstract

**Simple Summary:**

There are many burdens that cancer patients face during their illness: medical, organizational, social, psychological, spiritual, and so on. Often, patients cannot express all their needs, and on the other hand, clinicians do not pay enough attention to patients’ needs (other than medical needs). Therefore, a comprehensive assessment is needed for determining cancer patients’ non-medical needs. The Needs Evaluation Questionnaire (NEQ) is a simple, comprehensive, and easy-to-administer tool. Our study is the first to demonstrate an adaptation of the NEQ for Polish cancer patients during oncological treatment. The NEQ seems to be useful in assessing the unexpressed needs of cancer patients and in daily practice could improve the quality of patient–medical staff communication. Moreover, the NEQ may also be used in non-cancer patients.

**Abstract:**

Background: Cancer influences various aspects of patients’ functioning. Cancer patients face not only medical problems but also organizational, socio-psychological, and spiritual problems. Their needs often seem to be unrecognized because patients do not express their concerns and clinicians do not ask appropriate questions. Unmet needs impact patients’ quality of life. The aim of this study was to select, adapt, validate, and introduce a simple instrument for estimating cancer patients’ unmet needs in Poland. Methods: The Needs Evaluation Questionnaire (NEQ) was chosen for validation in a Polish population. The Polish version of the NEQ was developed with a back-translation procedure, as approved by a psycho-oncologist and a public health specialist. The psychometric properties of the NEQ (content analysis, reliability, construct validity, comprehensibility, and acceptability) were measured. Results: This study was performed on a group of 121 cancer patients. The median time of completion for the NEQ was 10 min. The form, length, and font size of the NEQ were accepted by the respondents. Overall, the meaning of the questions was well understood, with only a few cases of discreetly heterogeneous interpretation of the content. The questionnaire showed good reliability and internal factor structure validity. Conclusion: The NEQ is a simple, easy-to-administer instrument with good psychometric properties and seems to be useful in assessing the unexpressed needs of cancer patients.

## 1. Introduction

Cancer modifies the natural order of people’s needs related both to health and to general functioning. Patients lose previous functional ability and the capacity to live their normal life [1]. Disease affects the routine daily life of patients and their relatives. Because patients are focused on treatment and cure, their perceptions of the problems of everyday life are changed [2]. Cancer diagnosis, treatment, and follow-up influence various aspects of daily life. Changes to relationships with family and friends, a decreased ability to work, loss of social function, and reduced participation in other activities such as hobbies seem to be significant issues during cancer treatment [2,3]. These changes have negative impacts on a patient’s position in society and their self-confidence [2]. After cancer diagnosis, people have to face numerous problems; these are not only medically related (with cancer itself, oncological therapy, and adverse side-effects) but are also organizational, social, psychological, and spiritual. Therefore, cancer patients express various supportive care needs [2,4,5,6,7]. Non-medical needs include also communication with clinicians and the healthcare system [8,9,10,11,12,13,14,15,16]. Receiving an honest and understandable prognosis is important for improving patients’ quality of life [17]. Multidisciplinary and other relevant support should be offered to increase patients’ awareness of the effects of cancer and associated therapies not only on patients’ health but also on their private and professional lives. In daily practice, the medical team usually does not have adequate time to devote to the non-medical needs of patients. Physicians and nurses have priorities aligned with patient health, whereas patients may consider other problems they encounter to be of equal priority. Patients’ priorities and needs are often not identified and addressed because many patients still do not express their non-medical concerns to their clinicians or other medical staff [18,19,20]. Our previous study showed that only 21% of cancer patients in Poland received psychological support from psychologists and 4% of patients received help from priests [21]. Only 7% of cancer patients received support from a social worker [21].

Unmet needs represent a gap between what a health system currently delivers and the expectations of patients, resulting in a negative impact on quality of life [17,22,23,24,25,26].

The non-medical problems of cancer patients are still underappreciated in Poland. One of the main goals of the new National Oncological Strategy (NOS) is improvement of quality of life for patients and their relatives [27]. According to the NOS, a questionnaire for assessing cancer patients’ satisfaction and needs should have been introduced by the end of 2022. However, this has not happened. Moreover, in Poland, until now, there has not existed a validated questionnaire for investigating the unmet needs of cancer patients during therapy.

An adequate needs assessment tool could open up a communication channel between patients and medical staff. On the one hand, it would be a source of information about patient’s needs for professionals. On the other hand, it would give patients the opportunity to express their unmet needs.

The assessment of patient needs is very challenging due to personal, cultural and traditional differences between patients and medical staff; therefore, an appropriate tool for estimating unmet needs should be introduced into the health system. The assessment of satisfaction with healthcare and the efficacy of medical and non-medical interventions are important for patient wellbeing [17]. Some instruments used to evaluate these needs in cancer patients have previously been developed and used in some countries. [17,24,28,29]. Rimmer et al. [29] conducted a systematic review of 30 studies regarding assessment of the unmet needs of cancer patients. There were 24 instruments identified. There was extensive heterogeneity reported in their development, content, and quality [29]. It is difficult to select the best tool for clinical use, and further investigations are needed. Identification of a proper tool will be valuable for daily practice, research, and public interventions and support for cancer patients in order to meet their various needs and improve their quality of life. It is desirable that a simple instrument for estimating cancer patients’ unmet needs is introduced in Poland.

The aim of this study was to choose the most useful needs assessment tool for Polish cancer patients. We decided to investigate the appropriate literature and carry out selection and cultural and linguistic adaptation of the chosen tool. The specific objective of this study was validation of the Needs Evaluation Questionnaire in a Polish population. Satisfactory results from validation could be an argument for introducing this tool into the Polish healthcare system.

## 2. Materials and Methods

### 2.1. Selection of an Instrument to Assess the Needs of Polish Cancer Patients

A systematic search was conducted in the PUBMED database. Two search terms were used: (1) ‘needs assessment cancer patients’ and (2) ‘unmet needs cancer patients’. In the list of publications from 1 January 2000 to 31 December 2019, 5277 records referenced search term (1) and 1806 records referenced search term (2). Duplicate articles were excluded. Finally, the search yielded 6480 publications (including 323 clinical trials, 409 clinical studies, 76 meta-analyses, and 1585 reviews). Two investigators critically appraised and quality-rated publications to determine their relevance. For a study to be excluded, both researchers had to agree; disagreements were resolved by consensus. Based on these reviews, 384 publications were deemed to be of adequate potential relevance to this study.

In the selected publications, several assessment instruments for evaluating needs in cancer patients were described. After full-text evaluation using various inclusion criteria (empirical studies about patients’ needs, detailed description of instruments, validity described in the methodology, reliability described in the methodology, and validation performed), 63 articles were chosen, and 42 assessment scales were identified. In the process of selecting an instrument to use on Polish population data, several comparative reviews of needs questionnaires were used [7,17,24,29]. The article from 2004 by Wen et al. [17] proved particularly useful. The authors conducted a review of instruments to assess patient needs and selected 24 tools from 43 articles. One of these assessment instruments was the Needs Evaluation Questionnaire (NEQ).

The NEQ was originally designed and validated by Tamburini et al. at the Psychology Unit of Instituto Nazionale Tumori in Italy [8,30,31]. In the PUBMED database, there were 14 studies conducted using the NEQ [8,15,30,31,32,33,34,35,36,37,38,39,40,41]. Validation of the NEQ as a clinical tool for identifying the needs of hospitalized cancer patients was performed [8,30,31,35], and the contributions concluded that using the NEQ is functional, of value, and to be recommended. The original questionnaire is in Italian. The initial assessment of the NEQ was performed by Tamburini et al. [31] on 493 cancer patients. The authors used the NEQ to assess patients’ needs but not to estimate patient satisfaction with healthcare. Patients’ needs associated with the relationship between the patient and medical staff were the most important. Patients indicated the need for “more information about my future condition” (74%), “having a better dialogue with clinicians” (57%), “more information about my diagnosis” (56%), “more information about the exams I am undergoing” (52%) and “more explanation of treatments” (51%) [31]. Chiesi et al. [35] showed that the NEQ is an appropriate instrument across gender, age, cancer type, and phase of disease. The NEQ was also used to evaluate the needs of cancer outpatients and highlight the importance of determining and assessing unmet needs among patients in ambulatory care [33]. Konstantinidis et al. [34] used the NEQ for determination of unmet supportive care needs amongst hematological cancer survivors in Greece. Gangeri et al. [38] used the NEQ to evaluate the non-medical needs of candidates for liver transplantation for cancer—77% of respondents identified a need for information about future conditions, and 50% of respondents needed more explanation of examinations and treatments. Scaratti et al. [39] used the NEQ not only to estimate patients’ needs but also to estimate those of their caregivers. In Poland, the NEQ was used once to assess the needs of hospice patients and their relatives [42]. This study showed that similar needs were expressed by both patients and their relatives, with the most common being “control of the patient’s symptoms”.

The authors applied to Instituto Nazionale Tumori in Milan, Italy for approval to use the NEQ in Poland. Consent for translation into Polish was obtained, as well as for validation followed by use of the NEQ in a Polish population.

### 2.2. Preparation of Polish Version of the NEQ

The English version of the NEQ was available in the paper by Tamburini et al. [8]. The Polish version of the NEQ was developed with a back-translation procedure. The English version of the NEQ was translated into a Polish version by two independent professional translators. The Polish version was then translated back into English by another translator. Discrepancies from this back-translation were discussed, and adjustments were made to the Polish NEQ translation. The translation retains meaning but is not literal. Polite forms of addressing respondents that are accepted in Poland were taken into account. Some questions were modified according to Polish social and cultural backgrounds (for example, the spiritual advisor in the Polish version was replaced by a priest). Therefore, a psycho-oncologist and a public health specialist were involved in the preparation of the final Polish version of the NEQ. Additionally, ten extra questions were used to collect clinical and socio-demographic data. The final Polish version of the NEQ was presented to the authors of original version of NEQ. Polish and English versions of the questionnaire are included in the Appendix A.

### 2.3. Assessment of the NEQ

#### 2.3.1. NEQ Comprehensibility and Acceptability

The NEQ’s comprehensibility and acceptability for patients were evaluated using a validation procedure questionnaire (VPQ) designed specifically for this study. This questionnaire consisted of 2 parts: (I) assessment of the time taken to complete the questionnaire using median and interquartile range (25–75% IQR) and (II) assessment of the comprehensibility and acceptability of the questionnaire. The second part included 9 main closed-ended questions with an additional option for responding to some questions related to the structure, form and clarity, comprehensibility, and suitability of the questionnaire. Polish and English versions of the questionnaire are presented as Appendix A.

#### 2.3.2. Content Analysis

To verify the understanding of the meaning of phrases used in the Polish version of the NEQ a structured interview was conducted. The five phrases examined were: “intimacy”, “social service”, “commiseration”, “support from relatives”, and “being involved in the therapeutic decision”. During interview by a psycho-oncologist each patient was asked to specify themself the meaning of these phrases. The answers were documented and then classified into different categories of interpretation by investigators who had not carried out the interview.

#### 2.3.3. Validation of the Polish Version of the NEQ

##### Reliability

The validation procedure included comparison of two sets of responses to questions from the same patient within a two-week interval. In the analysis of the repeatability of answers to the questions, the percentage agreement was calculated (i.e., the percentage of observations for which all ratings were the same). Cohen’s Kappa coefficient was used to assess the agreement of two measurements of a qualitative variable [43]. The asymptotic error of the Cohen’s Kappa coefficient was estimated. Criteria for compliance according to Cohen’s Kappa value were adopted according to Landis and Koch: 0–0.20 indicated slight agreement, 0.21–0.40 fair agreement, 0.41–0.60 moderate agreement, 0.61–0.80 substantial agreement, and 0.81–1.00 almost perfect agreement [44].

Cohen’s Kappa coefficient was estimated using SPSS 28.0 software (IBM Corp. Released 2020. IBM SPSS Statistics for Windows, Version 28.0. IBM Corp., Armonk, NY, USA) (accessed on 1 February 2022). A *p* value <0.05 was considered to be significant.

##### Construct Validity

Confirmatory factor analysis (CFA) was used to estimate the validity of separating the five correlated factors for the NEQ. The model was initially proposed by Annunziata et al. [30] and is composed of five factors that correspond to five areas of needs: informative needs, needs related to assistance/care, relational needs, needs for psycho-emotional support, and material needs. The model of confirmatory analysis was built on the assumptions of Jöreskog [45].

Assessment of fit was based on two relative fit indices: the Comparative Fit Index (CFI) and Parsimony Normed Fit Index (PNFI). CFI values close to 0.9 and PNFI values above 0.5 are usually considered satisfactory [46]. Additionally, the standardized root mean square residual (SRMR) was used. SRMR represents the square-root of the difference between the residuals of the sample covariance matrix and the hypothesized model. Values below 0.08 reflect good fit [47].

The concordance of responses for the distinguished groups of questions characterizing similar phenomena was characterized by the Cronbach’s alfa coefficient [48]. The recommended cutoff value of Cronbach’s coefficient indicating good reliability of the questions was at a level of 0.7 [49].

Calculation of Cronbach’s alpha indexes and confirmatory factor analysis were calculated on data of 121 patients using JASP version 0.16.3 software (JASP Team (2022) (accessed on 1 February 2022).

### 2.4. Patients

The procedure of selecting the needs assessment instrument was conducted between July 2020 and December 2020. The NEQ assessment procedure was carried out between March 2021 and February 2022 in two oncological centers in Poland (University Hospital in Zielona Gora and Zaglebiowskie Oncology Center in Dabrowa Gornicza). The inclusion criteria for patients to participate in the study were as follows: (1) age ≥ 18 years old; (2) confirmed cancer diagnosis; (3) oncology treatment ongoing or having finished no longer than 3 months previously (with radical or palliative intent but not end-of-life care); and (4) current hospitalization for at least 3 days or after at least one hospitalization due to oncological treatment up to 3 months ago. Current or recent hospitalization was considered because the NEQ assesses some aspects that are only available in Poland in hospital (for example, Polish cancer patients can receive psychological and spiritual support from psycho-oncologists and priests during hospitalization but not in outpatient clinics).

The NEQ was administered to a group of 130 patients who met the inclusion criteria. After filling out the main NEQ form supplemented by demographic data and clinical data, patients were asked to complete the VPQ and then were interviewed by a psycho-oncologist to verify their understanding of the meaning of chosen phrases from the NEQ.

The study protocol was approved by the Ethics Committee of the University of Warmia and Mazury in Olsztyn (No. 30/2020). Participation in the study was voluntary. All study participants were informed about the aim of the study. Informed consent was obtained from all subjects and a blank copy of the consent form is included in Appendix A.

## 3. Results

### 3.1. Patient Characteristics

Participation in the study was offered to 130 cancer patients. The analysis was performed on a group of 121 cancer patients who decided to take part in the study. Respondents were aged 30–88 years (median age: 66 years). There were 72 women (60%) and 49 men (40%). Most patients had graduated from secondary education (66%), lived in cities (68%), were pensioners (63.5%), and were married or in a stable informal relationship (63%). Some 83% of patients analyzed did not have a doctor amongst close family or friends. The most common cancer type was digestive system cancer and breast cancer (Table 1).

### 3.2. Willingness to Participate in the Survey

Participation in the study was offered to 130 cancer patients, and 121 patients decided to take part in the study (93%). All of them completed both forms; the NEQ and the VPQ. A total of 106 patients agreed to interview with a psycho-oncologist (88%).

### 3.3. Prevalence of Needs

A total of 14 of the 23 NEQ questions were completed without any missing responses. The percentages of missing responses for remaining 9 questions were low (1–2%).

Patients most frequently expressed the following economic and organizational needs: for better services from the hospital (60%) and for more information relating to economic insurance (45%). Patients also frequently indicated informational needs for more information about their future condition (51%) and more information about diagnosis (39%), examinations (39%), and treatment (38%). Patients also expressed a desire to speak with people who have had the same experience (41%). Responders indicated relational needs for clinicians to be more sincere (42%) and more reassuring (37%) and to have a better dialogue with their doctors (30%). The needs that were less frequently expressed concerned help with eating, dressing, and going to the bathroom (18%); better attention from nurses (19%); and speaking with a spiritual advisor (19%) (Table 2).

### 3.4. Comprehensibility and Acceptability of the Questionnaire

The comprehensibility and acceptability of the NEQ were evaluated using the validation procedure questionnaire (VPQ).

The median time taken (25–75% IQR) for completion of the questionnaire was 10 min (5–15 min) for the main NEQ and 5 min (3–5 min) for demographic data and clinical data.

All patients except two considered that the format of the NEQ was good (98.4%). The font size was big enough for 113 responders (93.4%). Some 81% of patients indicated that the questionnaire was sufficiently long. A total of 105 responders (86.8%) claimed that the questions were understandable.

Only 16.5% of patients indicated that there were some questions that were difficult for them to answer clearly, and only 8 respondents (6.6%) claimed that they did not want to provide an answer for one or more questions.

Overall, the NEQ completely encompassed the patients’ needs. Only 8.3% of patients (10 individuals) indicated that they wanted to express more about their needs. The majority of patients (79.3%) claimed that completing this questionnaire may facilitate better contact with the doctor/nurse/other staff. A few participants (6.6%) identified important needs that they had not recognized before participation in this study (Table 3).

### 3.5. Content Analysis

The understanding of the meaning of phrases used in the Polish version of the NEQ was assessed by a structured interview with a psycho-oncologist.

The respondents were asked about their understanding of the term “intimacy” from the question “I need better respect for my intimacy”. Patients generally understood the intended meaning of that phrase—76% of respondents gave the correct definition of “intimacy”. For 52% of respondents, “intimacy” means no interference with both physical and psychological privacy and respecting boundaries. However, 24% of patients related “intimacy” to respecting the law in the context of personal data. Moreover, 8% of patients could not describe the meaning of this term, and 16% of responders gave unclear answers.

The next phrase to be investigated was “social service” from the question “I need to have more economic insurance information (tickets, invalidity, etc.) in relation to my illness”. Most respondents (87%) mainly associated “social service” with financial support, 2% of patients associated “social service” with institutional support, and 2% of responders associated it with rehabilitation equipment. However, 9% of patients were not able to define “social service”.

Patients, in general, could not properly define the term “commiseration” from the statement “I need to receive less commiseration from other people”. Some patients associated “commiseration” with something positive (39%) and others with something negative (61%). Some 20% of patients perceived “commiseration” as an increase in interest and willingness to help, whereas 19% of patients associated “commiseration” with increased sensitivity to others. Some 20% of individuals associated “commiseration” with too much nosiness and excessive interference in someone’s life. In the opinion of 41% of patients, “commiseration” was excessive pitying of someone.

The phrase “support from relatives” from the statement “I need to be more reassured by my relatives” was well understood. A total of 49% of patients associated this with psychological help/assistance, whereas 28% of patients associated it with physical help in the form of shopping, transport, cooking, etc. Some 19% of responders interpreted “support from relatives” as general help and care, and 4% of patients could not precisely specify the meaning of the phrase “support from relatives”.

The expression “being involved in the therapeutic decision” from the statement “I need to be more involved in therapeutic choices” was exactly understood by 63% of responders. Some 34% of patients perceived “being involved in the therapeutic decision” as cooperation with the oncologist (talking/listening/discussing and consent to the doctor’s therapeutic suggestions). Moreover, 29% of patients understood the phrase “being involved in the therapeutic decision” as themselves having a real influence on the treatment decision, and 17% of patients associated “being involved in the therapeutic decision” only with signing a consent form. Unclear answers were given by 12% of patients, and 8% of patients said “I do not know”.

### 3.6. Reliability

The reliability analysis of the NEQ included a test–retest conducted on a group of 46 patients. The response consistency rate was calculated, and the results of the reliability analysis are reported in Table 4. The questionnaire showed good overall reliability. According to criteria set out by Landis and Koch [44], answers to 16 from the 23 questions showed almost perfect agreement, answers to 6 questions showed substantial agreement, and answers to 1 question were in moderate agreement.

### 3.7. Construct Validity

Confirmatory factor analysis of the selected model showed a good overall fit for the model (χ^2^ = 518.37; *p* < 0.001). The values of the estimated fit indices (CFI = 0.801; PNFI = 0.612; SRMR = 0.077) were very close to the values considered to be satisfactory for model fit.

All estimated standardized factor loadings ranged from 0.52 to 0.84 and were significantly different from zero at *p* < 0.001, confirming good levels of internal factor structure validity. In general, inter-correlations between pairs of distinguished groups were very high (>0.7). Only the levels of inter-correlations between informative needs and relational needs and between informative needs and need for psycho-emotional support were close to 0.7 (Figure 1).

Cronbach’s alpha indexes for the five factors were ≥0.7 and close to 0.7 (need for psycho-emotional support) showing good levels of internal consistency (Table 5).

## 4. Discussion

Cancer patients are faced with a lot of physical, psychological, social, and spiritual problems [2]. Patients require support in solving those difficulties; however, it is a challenge for patients to identify and express their problems and needs. People interacting with the patient—relatives, friends, and especially healthcare staff—also have difficulties in recognizing what may be required. The issue of unmet needs in cancer patients is very important, especially as they may be related to patient wellbeing [2,7]. A better understanding of patients’ medical and non-medical needs could be valuable in the design of clinical and psychosocial interventions to improve patient satisfaction, quality of life, and treatment adherence [17]. There is an essential need to create tools to evaluate patients’ difficulties and needs related to cancer and its treatment.

The aim of this study was to select, adapt, and validate a tool for evaluating unmet needs during treatment among Polish cancer patients. In Poland, there is no standardized instrument for assessing the needs of cancer patients. However, several studies have reported using a variety of instruments. Some systematic reviews have pointed out the strengths of different tools [7,17,24,29]. Rimmer et al. [29] recommended three instruments, including the NEQ, for clinical use. The NEQ is simple, comprehensive and easy to administer. The NEQ is a self-administered questionnaire, which makes it easier to use; it does not take up too much staff time and is inexpensive. The NEQ consists of 23 main dichotomous quantitative questions. This relatively small number of questions is convenient for patients. The dichotomous scale is easy to complete and convenient for interpretation. The strengths of this tool are its simple structure and scale, understandable content, and ease of administration. The NEQ has a wide range including different kinds of unmet need: informative needs, psycho-emotional needs, relational needs, material needs, and needs related to assistance/care. The questionnaire could be used in clinical practice and also in research [30]. Therefore, the NEQ was chosen for validation in a Polish population.

The NEQ seems very easy to administer. The median time for completing the Polish version of the NEQ was 10 min, which is comparable to the results obtained by Italian authors (completion time of 5–10 min) [31,33]. The questionnaire is relatively short and includes 23 questions with a dichotomous scale (yes/no). The designers of the NEQ tested a four-point verbal Likert scale (not at all/a little/much/very much), but they noticed that grading the level of needs using this scale was difficult for cancer patients [31]. Tamburini et al. [31] demonstrated that the NEQ with a dichotomous scale was accepted by patients. Participation in the study was offered to 130 Polish cancer patients, and 121 of them consented to complete the questionnaire. All 121 patients completed the NEQ. Some 61% of questions were completed without missing responses; 9 questions (39%) had some missing data, but only at a very low level (1–2%). Other authors have shown percentages of missing values for each item ranging from 0–3% [8,31]. This low level of missing responses indicates the acceptability and comprehensibility of the NEQ.

In the Polish study, most of the patients considered that the form of the questionnaire was good. Additionally, in the VPQ responders declared that the NEQ questions were understandable and that the length of questionnaire and size of the font were appropriate. A total of 83.5% of patients indicated that there were no unclear questions. There may be concerns that patients could understand some phrases in different ways; therefore, content analysis should be used to examine understanding of items within the NEQ to ensure patients properly understand what is being asked. Tamburini et al. [31] verified the meaning of the following terms: “future conditions”, “sincere clinicians”, “intimacy”, “commiserated”, “reassured”, and “involved”. Patients generally understood the meaning of the above phrases; however, their interpretations were not homogeneous; for example, the term “involved” was interpreted as “being informed” (38%), “taking part in decisions” (32%), “being listened to considered” (30%) [31]. In the current study, content analysis was also assessed among Polish cancer patients. The examined terms that could be unclear or ambiguous for cancer patients were “intimacy”, “social service”, “commiseration”, “support from relatives”, and “being involved in the therapeutic decision”. Patients generally understood the meanings of these phrases, but there were some differences in interpretations between patients. For example, half of respondents described “intimacy” as no interference with both physical and psychological privacy and respecting boundaries, but a quarter of patients understood “intimacy” as respecting the law in the context of personal data.

Studies showed that the psychometric properties of the NEQ, including reliability, structure validity, and internal consistency, result in the NEQ being a suitable instrument for assessing patients’ needs [8,30,31,33]. In the current study, the test–retest procedure demonstrated a high level of reliability except for item 9. However, the response to Q9, “I need my symptoms (pain, nausea, insomnia, etc.) to be better controlled”, could change over time. In the present study, the structure validity and internal consistency were assessed using confirmatory factor analysis. CFA demonstrated a good overall fit for five correlated factors. Our findings were similar to analyses reported in previous studies [30,33]. Annunziata et al. [30] proposed division of NEQ items into five correlated subgroups: informative needs, needs related to assistance/care, relational needs, needs for psycho-emotional support, and material needs. All estimated factor loadings were high among patients [30]. Bonacchi et al. [33] showed a good overall fit for the five correlated factors model among outpatients. In the current study in a Polish population, good levels of internal factor structure validity were confirmed.

### 4.1. Clinical Implications

The NEQ, as a simple questionnaire, gives patients the chance to express their unmet needs and also provides a valuable source of information for medical staff. In the current study, most of the patients considered that completing the NEQ could help develop better relationships with doctors, nurses, or other health workers. Completion of the questionnaire may cause staff to pay attention to certain needs of cancer patients. Some respondents identified needs that they had not previously been aware of. The NEQ seems to be a practical tool due to its potential use in research and in the identification of the real gap between services and support in the healthcare system and patient expectations. In daily clinical practice, the NEQ could highlight an individual cancer patient’s unmet needs and, by helping to fulfil patient expectations, may improve their overall wellbeing. It is believed that meeting the needs of cancer patients will improve their quality of life and satisfaction with medical care and influence their therapeutic decisions. The NEQ seems to be an appropriate instrument that should be introduced into daily clinical practice.

### 4.2. Study Limitations

This study has some limitations. The search was based on articles available in the PUBMED database. The study was carried out only in two oncological centers in Poland. The clinical characteristics of the patients were not analyzed except for type of cancer (for example, performance status, clinical stage of cancer, and type of oncological treatment). There was no measurement of patients’ mini-mental status. The values of the estimated fit indices of confirmatory factor analysis were very close to the values considered to be satisfactory for model fit. The reliability analysis could be limited because some unmet needs change over time [41].

## 5. Conclusions

The Polish version of the NEQ seems to be an appropriate instrument for assessing cancer patients’ unmet needs due to its good reliability, structure validity, comprehensibility, and acceptability. The NEQ is simple and easy to administer when assessing patient expectations in relation to various needs related to information and connection with assistance/care as well as relational, material, and psycho-emotional needs. It can be assumed that the Polish version of the NEQ may also be useful in non-cancer patients.

## Figures and Tables

**Figure 1 cancers-16-01451-f001:**
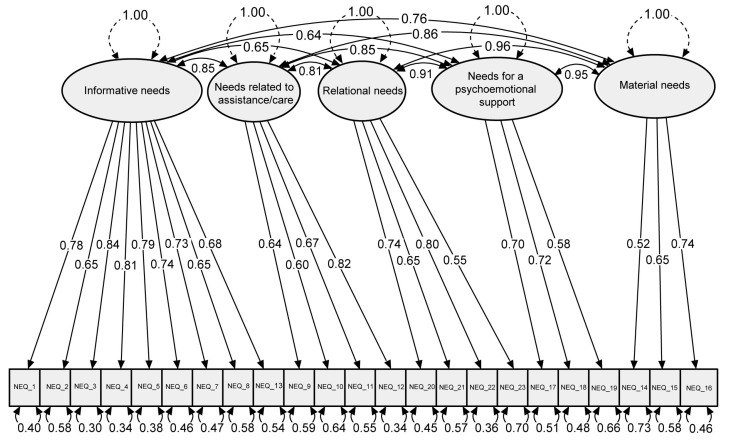
Standardized factor loadings and inter-correlations of the five-factor model.

**Table 1 cancers-16-01451-t001:** Characteristic groups.

		n	%
Gender			
	female	72	59.5
	male	49	40.5
Age (years) range; median (25–75% IQR)	30–88; 66 (55–70)		
Education			
	primary	14	11.6
	secondary	80	66.1
	high	27	22.3
Place of residence			
	city	82	67.8
	village	39	32.2
Professional activity			
	active	3	2.5
	active, but on sick leave	28	23.2
	unemployed	1	0.8
	pensioner	77	63.6
	disability pensioner	12	9.9
Marital status			
	married or in a stable informal relationship	76	62.8
	relationship broken during disease or in relation to disease	0	0
	single	8	6.6
	divorced	11	9.1
	widow/widower	26	21.5
Living with			
	partner	52	43.0
	partner and child/children	25	20.7
	child/children	10	8.3
	another family member	7	5.7
	partner and another family member	2	1.6
	alone	25	7
Having a medical doctor amongst close family or friends			
	yes	19	15.7
	no	101	83.5
	unknown	1	0.8
Cancer			
	head and neck	9	7.4
	upper digestive system	10	8.3
	lower digestive system	36	29.7
	lung	10	8.3
	breast	28	23.1
	gynecological	9	7.4
	prostate	14	11.6
	brain	2	1.7
	unknown	3	2.5

IQR—interquartile range.

**Table 2 cancers-16-01451-t002:** NEQ responses’ distribution.

			n	%
Q1	I need more information about my diagnosis	yes	47	38.8
		no	74	61.2
Q2	I need more information about my future condition	yes	62	51.2
		no	59	48.8
Q3	I need more information about the exams I am undergoing	yes	47	38.9
		no	73	60.3
		missing data	1	0.8
Q4	I need more explanations of treatments	yes	46	38.0
		no	74	61.2
		missing data	1	0.8
Q5	I need to be more involved in therapeutic choices	yes	35	28.9
		no	84	69.4
		missing data	2	1.7
Q6	I need clinicians and nurses to give me more comprehensible information	yes	46	38.0
		no	74	61.2
		missing data	1	0.8
Q7	I need clinicians to be more sincere with me	yes	51	42.1
		no	70	57.9
Q8	I need to have a better dialogue with clinicians	yes	36	29.7
		no	83	68.6
		missing data	2	1.7
Q9	I need my symptoms (pain, nausea, insomnia, etc.)to be better controlled	yes	45	37.2
		no	76	62.8
Q10	I need more help with eating, dressing, and going to the bathroom	yes	22	18.2
		no	98	81.0
		missing data	1	0.8
Q11	I need better respect for my intimacy	yes	28	23.1
		no	93	76.9
Q12	I need better attention from nurses	yes	23	19.0
		no	98	81.0
Q13	I need to be more reassured by the clinicians	yes	45	37.2
		no	76	62.8
Q14	I need better services from the hospital (bathrooms, meals, cleaning)	yes	72	59.5
		no	49	40.5
Q15	I need to have more economic insurance information (tickets, invalidity, etc.) in relation to my illness	yes	54	44.6
		no	66	54.6
		missing data	1	0.8
Q16	I need economic help	yes	33	27.3
		no	88	72.7
Q17	I need to speak with a psychologist	yes	30	24.8
		no	90	74.4
		missing data	1	0.8
Q18	I need to speak with a spiritual advisor	yes	23	19.0
		no	98	81.0
Q19	I need to speak with people who have had this same experience	yes	50	41.3
		no	71	58.7
Q20	I need to be more reassured by my relatives	yes	29	24.0
		no	92	76.0
Q21	I need to feel more useful within my family	yes	44	36.4
		no	76	62.8
		missing data	1	0.8
Q22	I need to feel less abandoned	yes	35	28.9
		no	86	71.1
Q23	I need to receive less commiseration from other people	yes	42	34.7
		no	79	65.3

**Table 3 cancers-16-01451-t003:** VPQ questions for the comprehensibility and acceptability of the NEQ.

	n	%
Is the form of the questionnaire good in your opinion?		
yes	119	98.4
no	1	0.8
no data	1	0.8
Is the font size big enough in your opinion?		
yes	113	93.4
no	7	5.8
no data	1	0.8
Do you think that the questionnaire is sufficiently long?		
yes	98	81.0
no, should be shorter	19	15.7
no, should be longer	4	3.3
Are the questions generally understandable in your opinion?		
yes	105	86.8
no	16	13.2
Are any questions difficult for you to answer clearly?		
yes	20	16.5
no	100	82.7
no data	1	0.8
Are there any questions you do not want to answer?		
yes	8	6.6
no	112	92.6
no data	1	0.8
Is there anything else you would like to discuss about your needs?		
yes	10	8.3
no	109	90.0
no data	2	1.7
Do you think that completing this questionnaire may facilitate better contact with the doctor/nurse/other staff?		
yes	96	79.3
no	25	20.7
Did you identify any important needs that you did not recognize before the questionnaire?		
yes	8	6.6
no	112	92.6
no data	1	0.8

**Table 4 cancers-16-01451-t004:** Reliability analysis.

Item Number	% of Consensus Responses	Cohen’s Kappa	Asymptotic Error of Kappa	*p*
Q1	93.5	0.78 **	0.12	<0.001
Q2	91.3	0.80 **	0.09	<0.001
Q3	93.3	0.83 *	0.09	<0.001
Q4	91.3	0.70 **	0.14	<0.001
Q5	97.7	0.93 *	0.07	<0.001
Q6	91.1	0.72 **	0.13	<0.001
Q7	95.7	0.90 *	0.07	<0.001
Q8	95.6	0.86 *	0.10	<0.001
Q9	87.0	0.60 ***	0.14	<0.001
Q10	97.8	0.85 *	0.15	<0.001
Q11	95.7	0.81 *	0.13	<0.001
Q12	93.5	0.69 **	0.17	<0.001
Q13	93.5	0.83 *	0.10	<0.001
Q14	91.3	0.81 *	0.09	<0.001
Q15	97.8	0.95 *	0.05	<0.001
Q16	97.8	0.88 *	0.12	<0.001
Q17	100.0	1.00 *	0.00	<0.001
Q18	97.8	0.90 *	0.10	<0.001
Q19	95.7	0.83 *	0.10	<0.001
Q20	95.7	0.86 *	0.10	<0.001
Q21	88.9	0.64 **	0.15	<0.001
Q22	95.7	0.83 *	0.12	<0.001
Q23	100.0	1.00 *	0.00	<0.001

* Almost perfect agreement; ** Substantial agreement; and *** Moderate agreement according to Cohen’s Kappa coefficient.

**Table 5 cancers-16-01451-t005:** Reliability correlation coefficient.

	Item Number	Cronbach’s α	95% CI
Informative needs	Q1–Q8, Q13	0.91	0.88–0.93
Needs related to assistance/care	Q9–Q12	0.74	0.65–0.80
Material needs	Q14–Q16	0.70	0.59–0.78
Need for psycho-emotional support	Q17–Q19	0.65	0.52–0.74
Relational needs	Q20–Q23	0.78	0.70–0.83

CI—Confidence interval.

## Data Availability

Data are contained within the article.

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
