# Peer review of "Validation of the Standardized Needs Evaluation Questionnaire in Polish Cancer Patients"

_cancers, 2024, doi:10.3390/cancers16081451_

Round 1

Reviewer 1 Report

Comments and Suggestions for Authors

 Osowiecka et al presented research on validation of the standardized Needs Evaluation Questionnaire in Polish cancer patients. The survey represents the needs Evaluation Questionnaire (NEQ) in Polish population. This is a specific objective-based survey and it is important at this point of time. The psychometric properties of the NEQ were evaluated for the small group of 121 patients.  Authors concluded that NEQ is a simple and good psychometric properties and seems to be useful in assessing the unexpressed needs of cancer patients. The manuscript also highlighted the study limitation. The manuscript can be accepted in its present form 

Author Response

Thank you very much for your review and nice comments.

Reviewer 2 Report

Comments and Suggestions for Authors

Comments:

Introduction

1. The introduction lacks a clear and concise thesis statement that outlines the main purpose or argument of the study. Adding a sentence that clearly states the aim or objective of the study would improve clarity and help the reader understand the focus of the research.

2. The introduction makes broad generalizations without providing specific evidence or examples to support the claims made. For example, statements like "Cancer modifies the natural order of people’s needs" and "Poor communication has also negatively impacted on patients’ quality of life" lack specific evidence or context to support them. Providing specific examples or evidence would strengthen these assertions.

3. The transition between discussing the need for multidisciplinary support and the discussion on unmet needs assessment tools is somewhat abrupt. Adding a sentence or phrase to bridge this transition and clarify the connection between these topics would improve the flow of the introduction.

4.  The introduction mentions the lack of a validated questionnaire to investigate unmet needs of cancer patients during therapy in Poland but does not provide an explanation of why this is important or how it relates to the study's objective. Providing a brief explanation of the significance of having a validated questionnaire would help justify the study's focus.

Method:

1.  While the section mentions a systematic search conducted in the PUBMED database, it lacks details about the specific search terms used, inclusion and exclusion criteria, and any additional databases searched. Providing these details would enhance the transparency and reproducibility of the study.

2. The section provides information about the selection of the Needs Evaluation Questionnaire (NEQ) but lacks a detailed rationale for why this instrument was chosen over others. Including a discussion of the specific criteria considered during the selection process and how the NEQ met those criteria would strengthen the justification for its selection.

3. Although the section briefly mentions the validation process of the NEQ for use in the Polish population, it lacks specific details about the validation methods employed. Providing information about the steps taken to validate the questionnaire, such as pilot testing, reliability analysis, and validity assessment, would enhance the rigor of the validation process.

4. While the section mentions the translation of the NEQ into Polish, it does not discuss any cultural adaptations made to ensure the questionnaire's relevance and appropriateness for the Polish population. Including a brief discussion of any cultural adaptations made during the translation process would provide valuable context for the questionnaire's use in the Polish context.

Comments on the Quality of English Language

Some sentences are lengthy and complex, which can make them difficult to understand. Consider breaking them down into shorter, clearer sentences to improve readability.

Author Response

Thank you very much for your comments. The suggestions are really valuable for improving the manuscript.

We would like to explain as follows:

Introduction

Introduction section was modified according to suggestions. The precise explanation of the aim of the study was done.

Method

Method section was reedited due to comments. Information about a systematic search was expanded. The decision of selection of the NEQ was explained in details. The information about cultural adaptation of the NEQ during translation into Polish was added.

Reviewer 3 Report

Comments and Suggestions for Authors

Dear authors. 

I have reviewed manuscript submitted to “Cancers” journal: "Validation of the standardized needs evaluation questionnaire in Polish cancer patients”.  

I think the title, abstract, summary, introduction, methods, results and conclusions, which include research limitations and clinical implications, are well written. The study focuses on the validation of the NEQ scale. The methodology is appropriate. The results are well developed and clear. The conclusions are also appropriate and fully consistent with the results presented. Study was approved by Ethical Committee. 

I only have a few comments and suggestions: 

  1. An essential aspect of the translation validation of an assessment tool is the semantic interpretation of the questions, taking into account the linguistic and cultural context. This study confirms that most of the questions of this tool were well understood, but not all. Of the 23 questions asked, 5 were interpreted in different ways. For this reason, the sentence in the abstract that says 'The meaning of the questions was well understood' is not entirely correct. I suggest change for a sentence like this: “Overall, the meaning of the questions was well understood, with only a few cases of discreetly heterogeneous interpretation of the content.” 

  1. Only patients who were currently or had recently been hospitalised were included. These patients are likely to have had a greater emotional impact than patients who did not need to be admitted to hospital because they were in a better clinical condition. Could the exclusion of outpatients be a potential source of selection bias? Can you explain why only patients who were recently or currently hospitalised were included? 

  1. If there were, it would be interesting to expand the data on the clinical characteristics of the patients. What is the evolutionary status of the patients enrolled in the trial? Are they at first diagnosis, advanced or metastatic, at the end of life, receiving oncological treatment or only palliative care? 

Sincerely.

Author Response

Thank you very much for your comments. The suggestions are really valuable for improving the manuscript.

We would like to explain as follows:

Ad.1 It’s true that strong statement that the meaning of the questions was well-understood should not be underlined. Thank you for your suggestion – we made changes in the text.

Ad.2 We decided to enroll to the study patients who were currently or had recently been hospitalized because the NEQ assesses some aspects, that are only available in Poland in hospital (for example Polish cancer patients can receive psychological and spiritual support from psycho-oncologists and priests during hospitalization but not in outpatient clinic).

Ad.3 The study included patients undergoing oncological treatment, both with radical and palliative intent. However, patients receiving palliative and end-of-life care were not included in the study. Clinical data were not collected, except for the type of cancer.